# Assessment of Serum Zn, Cu, Mn, and Fe Concentration in Women with Endometrial Cancer and Different Endometrial Pathologies

**DOI:** 10.3390/nu15163605

**Published:** 2023-08-17

**Authors:** Kaja Michalczyk, Patrycja Kapczuk, Patrycja Kupnicka, Grzegorz Witczak, Barbara Michalczyk, Mateusz Bosiacki, Dariusz Chlubek, Aneta Cymbaluk-Płoska

**Affiliations:** 1Department of Gynecological Surgery and Gynecological Oncology of Adults and Adolescents, Pomeranian Medical University, 70-111 Szczecin, Poland; 2Department of Biochemistry and Medical Chemistry, Pomeranian Medical University, Powstańców Wlkp. 72, 70-111 Szczecin, Poland; 3Department of Neonatology and Neonatal Intensive Care Subunit, ul. Dekerta 1, 66-400 Gorzow Wielkopolski, Poland; 4Department of Functional Diagnostics and Physical Medicine, Pomeranian Medical University, 70-111 Szczecin, Poland; 5Department of Reconstructive Surgery and Gynecological Oncology, Pomeranian Medical University, Al. Powstańców Wielkopolskich 72, 70-111 Szczecin, Poland

**Keywords:** endometrial cancer, uterine cancer, zinc, copper, iron, manganese, micronutrients

## Abstract

Background: There is conflicting evidence on the effect of specific micronutrient concentration and cancer risk. In this study, we investigated the differences in serum zinc, copper, iron, and manganese levels and different endometrial pathologies, including endometrial cancer. Methods: 110 patients with a confirmed diagnosis of endometrial cancer, benign uterine conditions (endometrial polyp, endometrial hyperplasia, uterine myoma), or normal endometrium were included in the study and assessed in terms of endometrial cancer risk factors. The measurements of serum micronutrients were conducted using inductively coupled plasma optical emission spectrometry. Results: When assessing for differences between serum concentrations of trace metals, we found significant differences in the distribution of Mn (*p* < 0.001) and Fe (0.034). There was also a significant difference in Cu/Zn ratio between the analyzed groups (*p* = 0.002). Patients’ BMI was found to influence Cu concentration, with obese patients having higher mean copper concentration (*p* = 0.006). Also, patients’ menopausal status was shown to influence Cu concentration with postmenopausal patients having higher Cu levels (*p* = 0.001). The menopausal status was found to influence Cu/Zn ratio (*p* = 0.002). Univariable regression analysis did not confirm that any of the micronutrients significantly influence the risk of endometrial cancer. Conclusion: The concentration of specific trace metals varies between different histopathological diagnoses of endometrial pathologies. Menopausal status and patient BMI are endometrial cancer risk factors impacted by the concentrations of Cu and Zn and their ratio.

## 1. Introduction

Numerous studies have shown correlations between specific trace metal concentrations and an increased risk of cancer development. However, there is still limited research and conflicting evidence on the relationship between specific trace elements and gynecological pathologies. 

Endometrial cancer (EC) is the most common gynecological malignancy, and its incidence is on the rise [1]. The widely established risk factors are increased patient age, obesity, increased BMI, insulin resistance, diabetes, and hormonal imbalance [2]. Moreover, patients’ history of early menstruation onset, late menopause, polycystic ovarian syndrome, infertility, presence of anovulatory cycles, and use of unopposed estrogen hormone replacement therapy or tamoxifen therapy also favor the development of endometrial cancer [3]. Genetic conditions such as Lynch syndrome, Cowden syndrome, and family history of endometrial or colorectal cancer also increase the risk of EC. Endometrial hyperplasia is a benign condition; however, it may act as a precursor state for endometrial cancer. It is usually correlated with presence of unopposed estrogen levels, which in absence of progesterone lead to abnormal proliferation of the endometrial tissue [4]. For many years, endometrial cancer was divided into two subtypes based on Bokhman classification, which divided the populations of EC patients based on EC tissue estrogen expression into Type 1 (estrogen-dependent) and Type II (estrogen-independent). Only in the last few years, a new molecular classification was proposed based on the findings of The Cancer Genome Atlas project (TCGA) [5]. The results of clinical trials PORTEC 3 and 4 showed the use of the new molecular classification in patient stratification based on clinical risk factors and molecular characteristics (POLE mutation, mismatch repair-deficient, p53 abnormal, and no specific molecular profile) [6,7,8]. Still, there are no specific biomarkers for endometrial cancer diagnosis. Some potential markers were identified (i.e., PTEN, TP53, POLE, and KRAS mutations); however, they lack sensitivity and specificity.

Zinc (Zn) and copper (Cu) are essential micronutrients that participate in multiple processes that regulate the functioning of the human body, including cellular metabolism and enzymatic activities. They also participate in oxidative stress processes and regulate formation of free radicals [9]. The homeostasis of Zn and Cu is strictly regulated by various mechanisms, and their imbalance may result in absorption competition resulting in impairment of their antioxidant properties. Important enzymes that require Cu for their function include Cu/Zn superoxide dismutase, ceruloplasmin, tyrosinase, and cytochrome oxidase. Cu concentration can be also affected by iron (Fe) concentration as increased Fe dietary intake was found to decrease serum Cu concentration [10]. Other micronutrients significant for human health include manganese, iron, copper and selenium, which have specific biological activities that participate in redox reactions, due to the presence of unpaired electrons in their atoms.

Altered levels of trace metals were found in different pathologies including immunological, degenerative, and inflammatory diseases [11]. As chronic inflammation and oxidative stress may induce microenvironment remodeling and carcinogenesis, we decided to evaluate the relationship between the levels of selected trace elements in women diagnosed with different endometrial pathologies. Despite several attempts to determine the roles of selected elements in endometrial cancer, the results remain inconclusive.

The International Agency for Research on Cancer (IARC) has created a classification to evaluate the carcinogenicity of specific agents to humans. The agents were classified based on scientific evidence from experimental animal studies and mechanistic or other relevant data. Four groups were identified:-Group 1—carcinogenic to humans;-Group 2A—probably carcinogenic to humans;-Group 2B—possibly carcinogenic to humans;-Group 3—not classifiable as to its carcinogenicity to humans.

Of the analyzed trace metals, the following were included in the classification: cadmium group 1 carcinogen; occupational exposure during iron and steel founding group 1 carcinogen; implanted foreign bodies of metallic cobalt, nickel, and alloy powder containing nickel, chromium, and iron group 2B carcinogen; iron sorbitol–citric acid complex group 3; iron–dextrin complex group 3. The Department of Health and Human Services (DHHS) as well as the IARC have not classified zinc for carcinogenicity due to the incomplete information from human and animal studies [12]. There are also no data on manganese carcinogenicity in IARC classification [13].

## 2. Materials and Methods

A total of 110 patients were included in a single-center study conducted at the Department of Gynecological Surgery and Gynecological Oncology of Adults and Adolescents, Pomeranian Medical University. For the purpose of the study, we have included patients with a confirmed diagnosis of endometrial cancer and benign uterine conditions (endometrial polyp, endometrial hyperplasia, and uterine myoma) or normal endometrium. All patients had a confirmed histopathological diagnosis. The exclusion criteria included recurrence of endometrial cancer, any previous form of cancer treatment, and presence of unbalanced/untreated chronic diseases. Patients with lost or incomplete data were removed from the study group before any statistical analysis. The study was conducted according to the guidelines of the Declaration of Helsinki and approved by the Ethics Committee of Pomeranian Medical University in Szczecin (protocol code KB-0012/27/2020 of 9 March 2020). Informed consent to participate in the study was obtained from all the patients.

### 2.1. Sample Collection and Storage

Two 5 mL venous blood samples were collected for trace element determination into 2 S-Monovette EDTA probes (Sarstedt, Germany). The samples were taken at the time of hospital admission for surgical procedure (hysteroscopy/laparoscopy or laparotomy depending on patient’s medical condition). One of the samples was then centrifuged for 15 min at a room temperature at 3000 RPM with a centrifugal force of 704× *g*. The samples were then stored at a temperature of −80 °C.

### 2.2. Measurement Methodology

The determination of trace metal concentration was performed using inductively coupled plasma optical emission spectrometry (ICP-OES, ICAP 7400 Duo, Thermo Scientific, Waltham, MA, USA) equipped with a concentric nebulizer and cyclonic spray chamber. Serum and whole blood samples underwent a microwave decomposition procedure with a microwave digestion system. Then, 65% HNO_3_ was added to the samples after defrosting and sample preparation. The specimens were then transferred into Teflon vessels and placed in the microwave digestion system MARS 5 (CEM, Matthews, NC, USA). The process of sample digestion was composed of two stages: an initial stage of 15 min, at which the samples were gradually heated up to 180 °C, and the second stage of 20 min at 180 °C. The protocol assumed a further 20-fold dilution of the digested samples. An amount of 500 μL of yttrium was added to the final standard sample concentration at 0.5 mg/L and 1 mL of 1% Triton (Triton X-100, Sigma-Aldrich, Poland). The samples were further diluted with 0.075% HNO_3_ (Suprapur, Merck, Poland) up to the volume of 10 mL and finally stored at 4–8 °C until analysis. The calibration curve was constructed using multielement standard solutions (ICP multielement standard solutions IV, IX, and XVI, Merck, Kenilworth, NJ, USA).

### 2.3. Statistical Analysis

Patients qualified for the study were assessed in terms of trace metal concentration (divided into two groups with increasing concentration), age, menopausal status, BMI, smoking, type 2 diabetes, and hypothyroidism. Patients were also differentiated based on their histopathological diagnosis (endometrial cancer, uterine fibroma, endometrial polyp, and normal endometrium). The objective of the analysis was to compare trace metal concentrations between different endometrial pathologies and to assess the relationship between patients’ characteristics, trace metal concentrations, and the occurrence of endometrial cancer. Due to a limited population of endometrial cancer patients, we did not perform a division between endometrial cancer staging based on FIGO classification. Study population characteristics are detailed in Table 1.

The normality of the groups was checked using the Shapiro–Wilk Test. In normally distributed groups, with only one categorical variable at a time, an ANOVA test was used to determine any differences between the average of the compared groups, with Tukey’s (Kramer’s) HSD (Honestly Significant Difference) to compare between means of specific pairs. For groups within which the distribution varied from normal, the Kruskal–Wallis test was used to assess differences in the dependent variable between the different groups. Multiple comparisons were conducted to compare the differences between pairs using the Post-Hoc Dunn’s test with the Bonferroni correction. In all the analyses, outliners were removed from the statistical analysis. To compare the differences between two independent groups, with an ordinal dependent variable, the Mann–Whitney U test was used as most of the data was not normally distributed, and outliners were present. The Mann–Whitney U test is more robust to the presence of outliers than the *t*-test. In order to assess the odds ratios (ORs) and the corresponding confidence intervals (95% CI) for each trace metal, they were calculated using univariate logistic regression models. *p*-value < 0.05 was adopted as the statistical significance threshold. Statistical analysis was performed with Statistica 10, StataSoft, Poland and R Statistical Analysis Software, R Foundation for Statistical Computing, Vienna, Austria.

## 3. Results

### 3.1. Group Characteristics

In the whole study population, the median patient age was 52 years of age, and median BMI was 27.06 kg/m^2^. For the purpose of the research, the patients were divided into four subgroups based on their histopathological diagnosis (myoma, normal endometrium, endometrial polyp, and endometrial cancer). A comparison between selected variables and patient characteristics was performed to determine if there were any changes between the groups. The results are presented in Table 2.

#### 3.1.1. Trace Metal Concentration

When assessing for differences between serum concentrations of Zn and Cu, the differences between the averages of all groups were not statistically significant (*p* = 0.714 and *p* = 0.328, respectively). The highest median serum Zn levels were found among patients diagnosed with uterine myoma (1.009 mg/L), while endometrial cancer patients were found to have the lowest median concentration (0.938 mg/L). As for cadmium, the lowest median values were noted for patients diagnosed with endometrial polyps (0.823). The distributions between specific subgroups are presented in Figure 1A,B.

Having calculated Cu/Zn ratio, the difference between the averages of some groups were big enough to be statistically significant (*p* = 0.018 with the corrected α using the Bonferroni correction method of 0.0083). The observed effect size f was medium (0.07). The Tukey–Kramer test showed significant differences between the means: myoma–endometrial polyp and myoma–normal endometrium with significantly higher mean Cu/Zn ratio in patients diagnosed with uterine myoma (*p* = 0.003 and *p* = 0.016, accordingly). The detailed results are presented in Figure 1C and Table A1, Appendix A.

The distribution of Mn concentration variated from normal. The Kruskal–Wallis H test indicated that there was a significant difference in the dependent variable between the different groups, χ2(3) = 16.33, *p* < 0.001, with a mean rank score of 55.43 for Group X1, 45.42 for Group X2, 29.89 for Group X3, and 35.47 for Group X4. The Post-Hoc Dunn’s test using a Bonferroni corrected alpha of 0.0083 indicated that the mean ranks of the following pairs are significantly different: X1–X3 and X1–X4 (see Appendix A, Table A1).

The averages of Fe concentration between the assessed groups were significantly different (*p* = 0.034), with 8.9% variance from the average. However, Tukey’s HSD showed no significant difference between the means of any pair. It is possible that a combined mean of more than one group differs significantly from the mean of one group or from the mean of other combined means. Specific results are listed in Appendix A, Table A2.

The distributions of specific trace metals between the specific subgroups are listed in Figure 1A–E.

#### 3.1.2. Patient Age

The Kruskal–Wallis test indicated that there is a significant difference in the dependent variable between the different groups, (*p* < 0.001), with a mean rank score of 34.5 for myoma, 51.5 for normal endometrium, 51.15 for endometrial polyp, and 82.5 for endometrial cancer. The Post-Hoc Dunn’s test using a Bonferroni corrected alpha of 0.0083 indicated that the mean ranks of the following pairs are significantly different: X1–X4 X2–X4 X3–X4 (Appendix A, Table A3).

#### 3.1.3. Patient Weight and BMI

The differences between the BMI averages of the groups were found to be statistically insignificant (*p* = 0.128). However, there are significant differences between the groups when accounting for patient BMI. The Kruskal–Wallis H test indicated that there is a significant difference in the dependent variable between the different groups, χ2(3) = 9.33, *p* = 0.025, with a mean rank score of 44.02 for myoma, 63.53 for normal endometrium, 46.34 for endometrial polyp, and 65.15 for endometrial cancer (Appendix A, Table A4).

### 3.2. Distribution of Trace Metal Concentration Based on Patient BMI

As a part of the study, an evaluation of the influence of patient BMI on trace metal concentration was conducted, regardless of histopathological results. Patients were divided into three subgroups based on their BMI: underweight and normal weight (BMI < 25), overweight (25–30), and obese (BMI ≥ 30). For all the variables, the differences between the groups were found to be insignificant, apart from Cu concentration (*p* = 0.006). The means were different between the groups of patients with BMI < 25 and BMI ≥ 30 (difference 0.197, *p* = 0.004). The specific details are presented in Table 3 and Appendix B, Table A5.

### 3.3. Analysis of Trace Metal Concentration Considering Patient Characteristics

Trace metal concentrations were analyzed based on different patients’ characteristics and presence of comorbidities. Significant differences were found between the values of Cu (*p* = 0.0013, f = 0.32) and Cu/ Zn ratio (*p* = 0.002, f = 0.31) based on patients’ menopausal status, with significantly higher values noted for patients after menopause. For other trace metals, the results were statistically insignificant. A significant difference was also found between Zn concentration based on presence of diabetes type 2. Zn concentrations of patients with diabetes type 2 were found to be significantly lower (*p* = 0.004, f = 0.30).

When assessed for presence of hypothyroidism and cigarette use, the differences between the concentrations of specific trace metals were not big enough to be statistically significant. Specific results are listed in Table 4.

### 3.4. Endometrial Cancer Risk Factor Analysis

The influence of endometrial cancer risk factors was conducted using univariate logistic regression. The results of the analysis are presented in Table 5. Menopause status, patient age, and type 2 diabetes were found to be statistically significant risk factors for the development of endometrial cancer.

## 4. Discussion

Only a few studies have been conducted trying to determine the correlation between trace metal levels in human serum and the risk of developing endometrial cancer and different endometrial pathologies. So far, there is conflicting evidence on the role of trace metals in different gynecological malignancies and pathologies. In this study, serum concentrations of zinc, copper, manganese, and iron were assessed. The obtained values were similar to the normal ranges suggested previously (see Table 6).

The levels of micronutrients may be altered due to dietary, lifestyle, or environmental factors caused by lifestyle habits, geographical location, and exposure to environmental contaminants including air, drinking water, and food. Imbalanced nutrition can cause dietary deficits of many important micronutrients. Epidemiological studies show a correlation between zinc deficiency and increased cancer risk [14,15]. In this study, lower median zinc levels were observed among patients diagnosed with endometrial cancer; however, the results were not statistically significant.

Copper is another important trace metal that regulates multiple metabolic and oxidative processes. Its levels are strictly regulated, and its imbalance can lead to the development and/or progression of inflammatory diseases and predisposition to carcinogenesis [11]. The role of Cu in malignancies has been widely studied. Elevated copper levels were found among serum and tissue specimens of patients diagnosed with breast, ovary, lung, and stomach cancers [16,17,18]. In this study, higher median levels of copper were noted among endometrial cancer patients, yet the results did not reach statistical significance.

The levels of zinc and copper are tightly regulated by compensatory mechanisms that maintain their concentrations within proper ranges. They may vary based on dietary intake and absorption efficiency, which occurs in the intestinal lumen through the absorptive cells. Increased zinc intake was demonstrated to lower copper absorption [19]. Low zinc levels and increased Cu may lead to increased oxidative stress and influence the function of multiple enzymes with antioxidant properties (including ceruloplasmin and Cu/Zn superoxide dismutase) [20]. An increased Cu/Zn ratio has been noted in studies conducted on different malignancies that included gastrointestinal [21,22,23], breast [24], ovarian [25], endometrial [26], and cervical cancers [27]. Our results showed significant differences between different histopathological diagnoses (*p* = 0.0021). There were statistically important differences between myoma and endometrial polyps, as well as between myoma and normal endometrial tissue, with patients diagnosed with uterine myoma having a higher median Cu/Zn ratio. The median serum Cu/Zn ratio was also higher in endometrial cancer patients when compared to patients with endometrial polyps or normal endometrial tissue; however, the differences were not big enough to reach statistical significance. Our findings show Cu concentration to be influenced by menopausal status and BMI index, as patients after menopause were found to have higher median Cu (*p* = 0.0013) and Cu/Zn levels (*p* = 0.0019). The concentration of copper was also found to vary between the groups of patients with different BMI indexes (*p* = 0.0061). A statistically significant difference was also found between the groups of patients with BMI < 25 and BMI ≥ 30 (difference 0.197, *p* = 0.0041).

Iron is another essential micronutrient, i.e., required for cellular respiration, energy metabolism, DNA replication and synthesis, and heme and iron–sulfur cluster production. On the other hand, due to its oxidative properties, it also participates in catalyzation and formation of toxic free radicals [28,29]. Excessive free radicals can lead to cell and tissue damage and thus to processes related to carcinogenesis. Cellular uptake and iron transport are tightly regulated to maintain proper concentrations of iron in its non-reactive forms in order to minimize oxidative stress and potential damage. The relationship between iron concentration and cancer risk has long been studied [30]. As the rates of proliferation and cell metabolism are generally higher in cancer cells than in normal tissue, their demand for iron is also significantly higher than normal cells, leading to greater oxidative stress. The higher demand for iron generates multiple changes in iron homeostasis, resulting in higher iron affinity, increased iron metabolism and input, inhibition of iron output, and finally iron accumulation [31]. The literature reports conflicting results from studies on the association between serum iron and cancer risk [32,33,34,35]. Among the studies that reported positive correlation, the risk varied by cancer type and patient gender [32,36]. Higher iron concentration has also been shown to be positively associated with the risk of diabetes and obesity, both risk factors for endometrial cancer [37,38]. Studies have evaluated dietary Fe intake and iron-related variables for the risk of endometrial cancer. A cohort study by Kabat et al. revealed no associations between meat, red meat, total dietary iron, iron from meat, heme, and non-heme iron intake and the risk of endometrial cancer [39]. On the other hand, a study conducted on a Swedish cohort of 60,895 patients revealed a modest positive association between heme iron, total iron, and liver intake and endometrial cancer risk, with no association for red and processed meat intake [40]. A study by Kallianpur et al. has also demonstrated an increased risk of EC for postmenopausal patients and among obese women with higher intake of heme iron [41]. In our study, significant differences were found between Fe concentration among the assessed groups of patients (*p* = 0.0342); however, upon a separate analysis between the specific subgroups of patients, the difference became insignificant. Patients with uterine myoma were found to have the highest serum Fe, while patients with normal endometrium or endometrial cancer tended to have a lower serum Fe concentration. Our findings show no influence between serum Fe and endometrial cancer risk (*p* = 0.466). A limitation of this study is that it did not take into consideration dietary micronutrient intake or supplement use. Patients did not complete questionnaires regarding their nutrition and eating habits or the use of dietary supplements. However, as all the assessed conditions (myoma, endometrial polyps, and endometrial cancer) are usually associated with abnormal and or excessive bleeding, we consider the group of patients homogenous for the possible nutrition intake requirements.

As a part of the study, serum Mn levels were also evaluated. Manganese is a cofactor of SOD dismutase that participates in oxidative stress reactions. A study by Tomczyk et al. has evaluated tissue concentration of Mn in patients with endometrial hyperplasia, endometrial polyps, endometrial cancer, and miscarriage. The authors found significantly higher concentrations of tissue Mn in EC patients compared to any other endometrial pathologies [42]. Up to that point, there had been no previous studies evaluating serum Mn concentration in patients diagnosed with different endometrial pathologies. Studies evaluating correlations between Mn concentration and cancer risk have shown no associations between Mn concentration and kidney or gastric cancer risk [43,44]. Our study revealed significant differences in manganese concentrations between the different groups of patients, specifically between patients diagnosed with myoma compared to endometrial polyps (*p* = 0.001) and myoma compared to endometrial cancer (*p* = 0.0526), with highest mean Mn values among patients with uterine myoma. Upon logistic regression, we did not find serum manganese concentration as a risk factor for endometrial cancer (*p* = 0.693).

The concentrations of trace metals can also be assessed in different tissue specimens or body fluids including urine samples and hair specimens. Urine levels of trace metals can sometimes be difficult to interpret as concentrations may vary throughout the day. They are highly dependent on patient diuresis and dietary intake. Increased urinary zinc elimination may be caused by intensive exercise, inflammation, high dietary intake, poor intestinal absorption, alcoholism, liver cirrhosis, diabetes mellitus, proteinuria, and starvation. A hair sample is another specimen that can be used to assess mineral concentration in the human body. Such samples can accurately reflect tissue storage and provide a long-term picture of trace metal content for up to multiple months prior to specimen collection. However, such trace metal concentration would not provide an assessment of current status and may be altered by multiple factors including patient age, gender, hair growth rate, and use of cosmetics and hair toners/colors. In this study, we have only assessed trace metal concentrations in patients’ serum. Further studies evaluating for possible differences in multiple specimens simultaneously would provide additional knowledge. The concentrations of metals measured in patient specimens may be altered not only by their dietary intake but also exposure.

For zinc, copper, iron, and manganese, the primary source of their intake is through diet. General exposure also occurs through the consumption of water, inhalation of air, and dermal contact with metal-containing products. There is also a risk of occupational exposure. People living in close proximity to or working in mining industries may be exposed to high levels of manganese dust or copper fumes through inhalation. Copper work exposure also includes agricultural work and work at facilities conducting copper processing. People working in coal mines, refining and smelting nonferrous metals, or living near waste sites may be exposed to high levels of zinc.

## 5. Conclusions

With growing evidence suggesting a relationship between altered serum trace metal concentration and endometrial cancer, further research on a larger group of patients is required to validate and evaluate our findings, especially with regard to serum copper and zinc. The relationship between zinc, copper, and its ratio and endometrial cancer seems to be complex and depends not only on the histopathological diagnosis but also on menopausal status and patient BMI, which are endometrial cancer risk factors. Studies trying to establish the optimal levels of serum trace metal levels should be conducted to determine their optimal concentrations. Dietary intake and supplementation should be evaluated as they may affect their concentrations.

## Figures and Tables

**Figure 1 nutrients-15-03605-f001:**
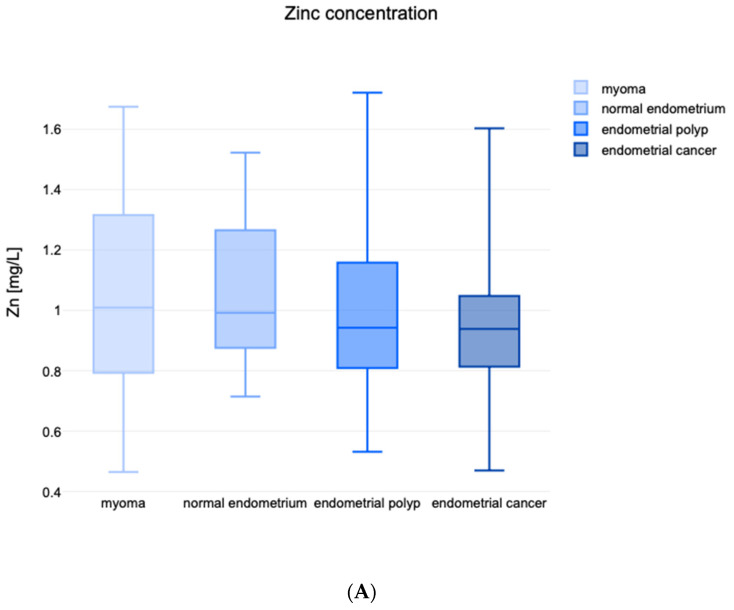
(**A**) Distribution of Zn. (**B**) Distribution of Cu. (**C**) Cu/Zn ratio. (**D**) Distribution of Mn. (**E**) Distribution of Fe.

**Table 1 nutrients-15-03605-t001:** Patients’ characteristics.

Characteristics	Overall
Age (years)	
<50	45
≥50–60	30
≥60	32
BMI (kg/m^2^)	
Normal (<25)	35
Overweight (≥25–30)	36
Obese (≥30)	25
Smoking	
Yes	7
No	101
Type 2 diabetes	
Yes	15
No	93
Menopause	
Yes	64
No	36
Hypothyroidism	
Yes	18
No	90
Histopathological diagnosis	
Endometrial cancer	21
Uterine fibroma	25
Endometrial polyp	48
Normal endometrium	16

**Table 2 nutrients-15-03605-t002:** Comparison of the selected variables between the study groups.

Variable	Myoma (X1)	Normal Endometrium (X2)	Endometrial Polyp (X3)	Endometrial Cancer (X4)	*p*-Value
Zn (mg/L)	1.001 (0.793; 1.316)	0.992 (0.876; 1.265)	0.942 (0.812; 1.265)	0.938 (0.814; 1.048)	0.714 *
Cu (mg/L)	0.923 (0.752; 1.003	0.887 (0.766; 1.055)	0.807 (0.699; 0.938)	0.915 (0.750; 1.148)	0.328 *
Cu/Zn ratio	1.133 (0.859; 1.482)	0.772 (0.713; 0.772)	0.862 (0.733; 0.862)	0.981 (0.805; 0.981)	0.018 *
Mn (mg/L)	0.009 (0.006; 0.012)	0.009 (0.002; 0.011)	0.004 (0.001; 0.006)	0.005 (0.002; 0.009)	<0.001 *
Fe (mg/L)	1.512 ± 0.607	1.107 ± 0.367	1.394 ± 0.433	1.192 ± 0.597	0.034
Age (years)	45 (40.5; 51.5)	52 (47; 55)	51 (43.5; 60.5)	70 (58.5; 79)	<0.001 *
Weight (kg)	69.8 ± 13.4	79.1 ± 15.0	71.1 ± 15.5	78.7 ± 21.2	0.128
BMI (kg/m^2^)	25.28 (23.32; 28.95)	29.72 (26.08; 34.18)	26.56 (21.78; 30.04)	30.27 (26.24; 37.52)	0.025 *

Data are reported as mean ± standard deviation or median (Q1; Q3). The groups were compared using ANOVA and Kruskal–Wallis test *. *p*-values < 0.005 were considered significant.

**Table 3 nutrients-15-03605-t003:** Differences in patient BMI between specific groups of patients.

BMI (kg/m^2^)	<25	25–30	≥30	*p*
Zn (mg/L)	0.957 ± 0.279	1.053 ± 0.229	1.075 ± 0.347	0.051
Cu (mg/L)	0.821 ± 0.184	0.896 ± 0.185	1.003 ± 0.334	0.006
Cu/Zn ratio	0.869 ± 0.238	0.976 ± 0.303	1.020 ± 0.376	0.127
Mn (mg/L)	0.005	0.006	0.007	0.676 *
Fe (mg/L)	1.357 ± 0.436	1.384 ± 0.481	1.317 ± 0.708	0.890

Data are reported as mean ± standard deviation or median. The groups were compared using ANOVA and Kruskal–Wallis test *. *p* values < 0.005 were considered significant.

**Table 4 nutrients-15-03605-t004:** Comparison of median trace metal concentrations based on patient characteristics and comorbidities.

	Zn (mg/L)	Cu (mg/L)	Cu/Zn ratio	Mn (mg/L)	Fe (mg/L)
Before menopause	0.999	0.795	0.843	0.007	1.441
After menopause	0.893	0.942	1.012	0.006	1.216
*p*	0.634	0.001	0.002	0.084	0.706
No diabetes	1.002	0.923	0.916	0.006	1.413
Diabetes type 2	0.856	0.840	0.939	0.006	1.043
*p*	0.004	0.340	0.785	0.633	0.051
No hypothyroidism	0.979	0.908	0.939	0.006	1.256
Hypothyroidism	1.086	0.868	0.853	0.008	1.552
*p*	0.098	0.173	0.083	0.187	0.577
Non-smoking	0.982	0.908	0.917	0.006	1.284
Smoking	1.151	0.822	0.928	0.010	1.734
*p*	0.174	0.623	0.369	0.255	0.468

The groups were compared using Mann–Whitney U test. *p* values < 0.005 were considered significant.

**Table 5 nutrients-15-03605-t005:** Univariate logistic regression.

Characteristics	OR	95%Cl	*p*-Value
Zn *	0.70	0.27–1.83	0.466
Cu *	1.31	0.49–3.49	0.587
Cu/Zn *	2.36	0.87–6.43	0.093
Mn *	0.81	0.29–2.30	0.693
Fe *	0.70	0.27–1.83	0.466
Age *	7.27	1.99–26.57	0.003
Weight *	1.36	0.51–3.68	0.540
BMI *	2.14	0.78–5.91	0.517
Menopause	20.45	12.63–159.0	0.004
Smoking	3.46	0.71–18.81	0.124
Diabetes type 2	14.91	4.296–51.75	<0.001
Hypothyroidism	0.21	0.03–1.64	0.136

* concentration > median.

**Table 6 nutrients-15-03605-t006:** Serum concentrations of selected trace metals.

	Normal Range	Obtained Results for Endometrial Cancer Patients
Zn	80–100 μg/dL	93.8 μg/dL (81.4; 104.8)
Cu	60–140 μg/dL	91.5 μg/dL (75.0; 114.8)
Mn	0.4–0.85 μg/L	0.5 μg/L (0.2; 0.9)
Fe	60–170 μg/dL	98.1 μg/dL (80.5; 98.1)

## Data Availability

The data presented in this study are available on request from the corresponding author. The data are not publicly available due to ethical restrictions.

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
