# Peer review of "Assessment of Serum Zn, Cu, Mn, and Fe Concentration in Women with Endometrial Cancer and Different Endometrial Pathologies"

_nutrients, 2023, doi:10.3390/nu15163605_

Round 1
Reviewer 1 Report
The authors have examined the relationship between a few serum trace element concentrations and the occurrence of endometrial cancer and some associated variables. The research appears to be well-performed, while the manuscript is generally well written.
My biggest concern is the improper use of significant figures. The average BMI is reported to 4 significant figures. For this to be possible, the height of the subjects had to be measured to 4 significant figures, which certainly was not done. Similarly, serum metal concentrations are reported to up to 5 significant figures. Do the authors really think they are measuring these levels to a precision of 5 sig. fig.'s? The standard deviations suggest that 3 is pushing things. This must be changed.
For the centrifugation, give "x g" or centrifuge model and speed.
Was informed consent written? If written, indicate in manuscript; if not, give ethics committee approved rational.
Avoid use of first person voice (e.g., "we") except in introduction and conclusion.
Manuscript would read better if so many sentences did not start "There are" or similar.
Numerous minor grammatical and related issues. Puncuation using "however" should be reviewed.
Author Response
My biggest concern is the improper use of significant figures. The average BMI is reported to 4 significant figures. For this to be possible, the height of the subjects had to be measured to 4 significant figures, which certainly was not done. Similarly, serum metal concentrations are reported to up to 5 significant figures. Do the authors really think they are measuring these levels to a precision of 5 sig. fig.'s? The standard deviations suggest that 3 is pushing things. This must be changed.
Thank you for your comments; we have improved the sig figs used in the manuscript
For the centrifugation, give "x g" or centrifuge model and speed.
We have added this information into the manuscript
Was informed consent written? If written, indicate in manuscript; if not, give ethics committee approved rational.
This was listed in the sample collection and storage, we have moved this information higher up in the methods section. Please see the lines 110-113
Avoid use of first person voice (e.g., "we") except in introduction and conclusion.
Manuscript would read better if so many sentences did not start "There are" or similar.
Numerous minor grammatical and related issues. Puncuation using "however" should be reviewed.
We have corrected the language used in the manuscript
Reviewer 2 Report
Since there is l conflicting evidence on the relationship between specific trace elements and gynecological pathologies the authors have investigated several trace metals in women with endometrial cancer and different endometrial pathologies. Overall, the study did not confirm any of the micronutrients to significantly influence the risk of endometrial cancer. But the authors conclude that zinc, copper and its ratio and endometrial cancer seems to be complex and depends not only on the histopathological diagnosis but also menopausal status and patients’ BMI, which are endometrial cancers’ risk factors. Since the study only included 110 patients including normal endometrium the authors propose further studies and to identify optimal serum trace metal levels and to better understand dietary intake and supplementation.
Although there is some interest to evaluate, whether trace metals have an impact on endometrial diseases, this preliminary study did not come to a firm conclusion so that additional studies are proposed. Such information does not worthy enough although the authors should be encouraged to perform such studies so that any conclusion can be based on a broader information.
Author Response
Dear reviewer, we would like to thank you for your comments
In our manuscript we have obtained the following results: we found significant differences in the distribution of Mn (p<0.001) and Fe (0.034). There was also a significant difference in Cu/Zn ratio between the analyzed groups (p=0.002). Patients’ BMI was found to influence Cu concentration, with obese patients having higher mean copper concentration (p=0.006). Also, patients’ menopausal status was shown to influence Cu concentration with postmenopausal patients having higher Cu levels (p=0.001). The menopausal status was found to influence Cu/Zn ratio (p=0.002). Univariable regression analysis did not confirm any of the micronutrients to significantly influence the risk of endometrial cancer. Conclusion: The concentration of specific trace metals varies between different histopathological diagnoses of endometrial pathologies. Menopausal status and patients’ BMI, endometrial cancers’ risk factors impact the concentrations of Cu, Zn and their ratio.
We agree that the study population sample was limited and only included 110 patients – this is why we conclude that further research is needed to evaluate our findings.
There are numerous studies that have been conducted on different populations of patients, however, so far, there is a very limited research investigating their evidence in gynecological pathologies.
We believe our study adds further evidence to the field
We have made some minor changes to the manuscript in accordance to the comments of other reviewers. Please see the improved version of the manuscript
Reviewer 3 Report
Interesting paper. However, there are few major points that need adressing. Examined elements are also influenced by outside burdening- professional exposure etc. which authors did not mention. Please adress them when You did not include them in Your research. Serum analysis is fine, although hair is maybe better to asses long term exposure.There is also plenty of published paper on refferal values for polish and other population giving values of Zn, Cu, Mn , Fe... in population. Please use them. Maybe You should put them in table, along with Your measured ones for comparison and put them in discussion. But these are all minor comments.
Author Response
Dear reviewer,
We would like to thank you for your comments. We have added some additional information regarding the professional exposure and the different tissue/ body fluids specimens in which trace metals can also be assessed. Please see the improved version of the manuscript
Ad 1.
We added the following information: For zinc, copper, iron and manganese, the primary source of its intake is through diet. The general exposure occurs also through the consumption of water, inhalation of air, and dermal contact with metal-containing products. There is also a risk of occupational exposure. People living in close proximity to or working in mining industries may be exposed to high levels of manganese dust or copper fumes through inhalation. Copper work exposure also includes agriculture work and facilities conducting copper processing. People working in coal mines, refining and smelting of nonferrous metals or living near waste sites may be exposed to high levels of zinc.
Ad 2.
We created Table 7 comparing the obtained values and the reference normal ranges
Please see the improved version of the manuscript
Round 2
Reviewer 2 Report
No further comments